# MOTION SCORE MATCHING IN VIDEO GENERATION

## ABSTRACT

Subtle motions in videos are critical for overall realism for video generation but are overlooked by current score distillation methods like distribution matching distillation. Current distillation methods prefer to match the style first, since it takes up most of the numerical significance. Such a distillation scheme will only create poorly generated motions, severely degrading the overall realism after distillation. To address this, we propose motion score and enforce the matching of motion distribution in distillation. We show that matching motion distribution is vital for the quality of generated videos.

## 1 INTRODUCTION

Diffusion models have revolutionized the area of visual generation. With proven scalability, they quickly become the driving algorithms of many visual creation platforms, spanning across image generation(Sohl-Dickstein et al. (2015); Ho et al. (2020); Song et al. (2020); Nichol & Dhariwal (2021); Ho & Salimans (2022); Karras et al. (2022)) and video generation(Ho et al. (2022b;a); Singer et al. (2022); Hong et al. (2022); He et al. (2023)). The quality of visual generation from diffusion models has improved unprecedentedly over previous generative models like VAEs(Kingma & Welling (2013)), GANs(Goodfellow et al. (2016)) and Normalizing Flows(Papamakarios et al. (2021)). One critical insight behind the success of diffusion models is the change from direct modeling on probabilities to the modeling of score function of distributions, which is proven to be more generalizable and scalable.

With the continuous emergence of foundational video diffusion models trained on enormous data like HunyuanVideo(Kong et al. (2024)), Wan2.1/2.2(Wan et al. (2025)), CogVideo(Yang et al. (2024); Hong et al. (2022)) etc., the modeling on common video distributions is reasonably well handled by these state-of-the-art models. But, as score models, they inevitably require tens to hundreds of forward passes on some large neural network to walk through the PF-ODE trajectory and reach a high probability region that produces a good quality video. This issue is more significant for video generation models as they are generally larger than image generation models. The computational and temporal cost limits the broader usage of these powerful models for interactive tasks and edge deployment. Reducing the number of sampling steps, but still maintaining the same level of generation quality, is a challenge for current video generation models.

To accelerate the sampling process, various approaches are proposed. Many methods focus on using fewer sampling points on the PF-ODE trajectories discovered by the original diffusion models(Song et al. (2020); Luhman & Luhman (2021); Chen et al. (2025); Salimans & Ho (2022)). Among which many apply the idea of Progressive Distillation(Salimans & Ho (2022)). Consistency Distillation(Song et al. (2023); Song & Dhariwal (2023); Kim et al. (2024)) is another stream of methods that do not exactly follow the original PF-ODE trajectories, but try to map any point from the trajectories to its clean origin. These two distillation schemes have successfully reduced the number of sampling step from hundreds to only handful or even only one, effectively reducing a large portion of the total computational expense. But both schemes need the student generator to mimic the full denoising trajectory at various locations, thus they either need full denoising trajectories or iterative teacher evaluations during training. The computational cost during training is high hence not ideal for distilling video diffusion models.

Distribution Matching Distillation(Yin et al. (2024b;a; 2025); Huang et al. (2025)) shows its promising future in diffusion distillation. This method trains the student generator to provide the same output as the original diffusion model on a distributional level. The student generator is not trained

to mimic the original PF-ODE trajectory, thus there is no need to simulate it during training. This reduces the computational cost spent on teacher evaluation during training, making it practical for distilling video diffusion models(Yin et al. (2025); Huang et al. (2025)).

Distribution Matching Distillation(DMD) does not enforce a strict matching on a trajectory level, but enforces a matching on the KL-divergence between the the student's generated distribution and the teacher's multi-step outputs estimated by their score functions. Though the design of this algorithm makes its extension from image diffusion to video diffusion practical, we found that the current matching on score functions are not ideal for videos due to the extra temporal dimension. The subtle movements that are vital for the overall realism in videos are usually buried in the diffused video clips when computing their score functions. After distillation, the generated videos possess good style similarity but poor motion quality.

We argue that the still appearance and motion in videos are equally important for generating realistic videos. The diffusion (adding noise) process in the computation of score functions will bury the subtle motion signal in large noise, making it very difficult for the student generator to learn correct motion behavior from the teacher. We present Motion Score Matching(MSM) to handle this issue by explicitly matching the motion related scores during distillation.

In summary, our contributions are

1. Identify two causes of buried motion signal during distribution matching of diffusion models.

2. Formulate motion score and use motion score matching to explicitly match motion distribution in videos, and prove that preserving motion signal is vital for video generation quality.

3. Extend our formulation of motion score to a dedicated foundational motion score model, and point out the benefits of building a foundational motion score model.

## 2 RELATED WORK

**Diffusion Models** Diffusion models have become a cornerstone in generative modeling for various modes including image(Ramesh et al. (2022); Rombach et al. (2022); Saharia et al. (2022)), audio(Kong et al. (2020)) and video(Ho et al. (2022a;b); Singer et al. (2022)). Introduced by Ho et al. (2020), these models learn to generate data by iteratively denoising Gaussian noise through a reverse diffusion process. This framework was further formalized using stochastic differential equations (SDEs), enabling continuous-time modeling of the diffusion process Song et al. (2021). Given the continuously increasing demand for generating longer videos with higher resolution, diffusion models shift to operating in a compressed latent space, achieving remarkable quality(Blattmann et al. (2023)).

**Accelerating Diffusion Inference** Typically, a diffusion model requires many denoising steps to provide high quality samples(Ho et al. (2020)). Given the original PF-ODE trajectory learned by the diffusion model, there is a series of works trying to reduce the number of sampling points along the trajectory. DDIM(Song et al. (2020)) uses deterministic samplers, reducing the sampling steps to around 50. More advanced numerical solvers like DPM solvers(Lu et al. (2022)) further pushes down to around 10. To further reduce the number of sampling steps, distillation on the original diffusion models is required. Progressive distillation iteratively halves the sampling steps by training student models on teacher trajectories Salimans & Ho (2022). Consistency distillation enforce self-consistency along probability flow ODEs, enabling one to few-step generation Song et al. (2023). Extensions like consistency trajectory models(Kim et al. (2024)) learn the full ODE trajectory for tractable sampling. Data-free approaches, such as BOOT(Gu et al. (2023)), distill models without access to training data by bootstrapping synthetic samples. Rectified flow techniques, as in InstaFlow(Liu et al. (2023)), straighten ODE paths to enable accurate one-step predictions.

**Distribution Matching in Generative Models** Distribution matching objectives have long been used in generative modeling to align synthetic and real data distributions without explicit pairwise supervision. Generative Adversarial Networks(Goodfellow et al. (2020)) pioneered this by training a discriminator to distinguish real from fake samples, providing gradients to the generator. This

adversarial approach has inspired hybrid losses in various domains, combining distribution matching with regression or perceptual losses for stability and quality.

**Distribution Matching Distillation** Building on these ideas, Distribution Matching Distillation (DMD) transforms multi-step diffusion models into efficient generators by matching the distributions of generated and teacher-denoised samples, augmented with regression losses. Initially developed for images to enable high-quality one-step synthesis(Yin et al. (2024b;a)). DMD has also been extended to video generation, distilling bidirectional diffusion transformers into few-step autoregressive models for fast, causal video synthesis Yin et al. (2025). But we found that the current DMD scheme leads to buried motion signal during the distillation process, making the distilled generator performs poorly on generating coherent and clear motion.

**Direct Motion Loss on Final Generated Videos** This is the most direct approach to improve the motion quality, used in Zhai et al. (2024). Traditional computer vision methods like optical flow, frequency disection and pretrained motion detection network is applied on the final generated videos to evaluate the motion quality. And the same procedure is applied on the teacher-generated videos to serve as the motion ground-truth samples. A loss between the student motion and the teacher motion is then applied to enforce the similarity on motion. But such methods fail when the student only generates the same output as the teacher on a distributional level, which means the student will give similar but not exactly the same videos. For example, given the same input prompt for text-to-video generation, the student can generate the same-style video, but the specific object, the moving direction are similar but not the same. Therefore optical flow methods should naturally provide different results and the loss is not meaningful. For distributional level distillation, it is not meaningful to enforce motion on final generated videos because the videos inherently do not have exactly the same motion.

## 3 BACKGROUND

This section introduces the formulation of diffusion models and distribution matching distillation to provide necessary grounds and notations, which will facilitate the introduction of our methods in the next section.

### 3.1 VIDEO DIFFUSION MODELS

Video diffusion models extend the principles of diffusion-based generative models from static images to dynamic video sequences, incorporating temporal dependencies to generate coherent motion. A video is typically represented as a tensor $\boldsymbol{x}_0 \in \mathbb{R}^{T \times H \times W \times C}$, where $T$ denotes the number of frames, $H$ and $W$ are the spatial dimensions, and $C$ is the number of channels (e.g., RGB). The fundamental setup involves a forward diffusion process that progressively adds Gaussian noise to the original video $\boldsymbol{x}_0$, transforming it into a noisy version $\boldsymbol{x}_t$ at timestep $t$, where $t = 1, \ldots, T_d$ and $T_d$ is the total number of diffusion steps. This process is Markovian, defined by the transition distribution

$$q(\boldsymbol{x}_t \mid \boldsymbol{x}_{t-1}) = \mathcal{N}(\boldsymbol{x}_t; \sqrt{1 - \beta_t}\boldsymbol{x}_{t-1}, \beta_t\boldsymbol{I}) \tag{1}$$

with $\beta_t$ being a predefined variance schedule that increases over time, ensuring that $\boldsymbol{x}_{T_d}$ approximates isotropic Gaussian noise.

The reverse process aims to recover the original video from noise by learning a parameterized model $\boldsymbol{p}_\theta(\boldsymbol{x}_{t-1} \mid \boldsymbol{x}_t)$ that approximates the true posterior $q(\boldsymbol{x}_{t-1} \mid \boldsymbol{x}_t)$. In practice, this is achieved by training a neural network, often a U-Net architecture augmented with temporal convolutions or attention mechanisms (e.g., 3D convolutions or spatio-temporal transformers), to predict the noise $\boldsymbol{\epsilon}$ added at each step. The training objective is derived from the variational lower bound on the negative log-likelihood, simplified to minimizing the mean squared error:

$$\nabla_\theta \mathcal{L}_{\text{Diffusion}} = \mathbb{E}_{t,\boldsymbol{x}_0,\epsilon}\left[||\epsilon - \epsilon_\theta(\boldsymbol{x}_t, t)||^2\right] \tag{2}$$

where $\epsilon \sim \mathcal{N}(\boldsymbol{0}, \boldsymbol{I})$, $\boldsymbol{x}_t = \sqrt{\bar{\alpha}_t}\boldsymbol{x}_0 + \sqrt{1 - \bar{\alpha}t}\epsilon$, and $\bar{\alpha}t = \prod_{s=1}^{t}(1 - \beta_s)$. This setup allows the model to denoise step-by-step, generating new videos by starting from pure noise $\boldsymbol{x}{T_d} \sim \mathcal{N}(\boldsymbol{0}, \boldsymbol{I})$ and iteratively applying the reverse transitions.

Moreover, advancements like latent diffusion models compress the video into a lower-dimensional latent space using a pre-trained video autoencoder, reducing computational overhead. The forward diffusion then operates in this latent space, with the final output decoded back to pixel space(Blattmann et al. (2023); Hong et al. (2022); Zhou et al. (2022)). This fundamental framework underpins models like Video Diffusion Models (Ho et al. (2022b)) and Make-A-Video(Singer et al. (2022)), enabling applications in video synthesis, editing, and super-resolution while maintaining spatio-temporal consistency.

## 3.2 DISTRIBUTION MATCHING DISTILLATION

Distribution matching distillation(DMD) is a method designed to reduce the number of sampling steps for diffusion models(Yin et al. (2024b;a)). Unlike other trajectory distillation methods which requires extentive computation on the trajectory simulation, DMD does not enforce the student generator to mimic the PF-ODE trajectories from the teacher, but to minimize the reverse KL divergence across randomly sampled timesteps $t$ between the smoothed data distribution $\boldsymbol{p}_{data}(\boldsymbol{x}_t)$ and the student generator's output distribution $\boldsymbol{p}_{gen}(\boldsymbol{x}_t)$. The gradient of this objective is approximated by the score functions on teacher and student generator.

$$\nabla_\phi \mathcal{L}_{\text{DMD}} \triangleq \mathbb{E}_t \left( \nabla_\phi \text{KL} \left( p_{\text{gen},t} | p_{\text{data},t} \right) \right) \tag{3}$$

$$\approx -\mathbb{E}_t \left( \int \left( s_{\text{data}} \left( \Psi \left( G_\phi(\epsilon), t \right), t \right) - s_{\text{gen},\xi} \left( \Psi \left( G_\phi(\epsilon), t \right), t \right) \right) \frac{dG_\phi(\epsilon)}{d\phi}, d\epsilon \right)$$

where $\Psi$ is the forward diffusing process as in Equation 1, $G$ is the efficient student generator we want to get with parameter $\phi$. $s_{\text{data}}$ is the score function trained on original raw data, which is the teacher diffusion in this case. Note that $s_{data}$ does not have a parameter variable, meaning that it is frozen during distillation. $s_{\text{gen},\xi}$ is the score function on the student generator, which is trained together with the student generator $G$ alternately to keep track of the evolving distribution on student.

## 4 METHODS

With video diffusion models distilled by DMD loss, we observed poor quality on the motion student generated videos. We identify two causes for poor motion quality first, and then give our approach to explicitly enforce the learning on motion distribution.

## 4.1 BURIED MOTION SIGNAL

The core of distribution matching is to enforce the student to match the score functions from the teacher diffusion model. Because basic assumption here is that the teacher model $s_{\text{data}}$ (Kong et al. (2024); Wan et al. (2025); Hong et al. (2022); Yang et al. (2024)) is trained with enormous scale of data, we believe the teacher provides a good modeling on the distribution of general videos given text or other conditional input. This means

$$s_{\text{data}}(\boldsymbol{x}|\boldsymbol{c}) \tag{4}$$

is assumed as accurate, where $\boldsymbol{x}$ is a video clip, and $\boldsymbol{c}$ is a text prompt. Splitting $\boldsymbol{x}$ into frames, we have

$$s_{\text{data}}\left( \left[ \boldsymbol{x}^1, \boldsymbol{x}^2, \boldsymbol{x}^3 \ldots \boldsymbol{x}^N \right] | \boldsymbol{c} \right) \tag{5}$$

where $\boldsymbol{x}^i$ is the $i$-th frame within the video clip $\boldsymbol{x}$. No matter the quality of video, DMD loss works by injecting noise into the video first and then compute its score functions in Equation 3. The actual score function is computed on

$$s_{\text{data}}\left( \left[ \alpha_t \boldsymbol{x}^1 + \sigma_t \epsilon, \alpha_t \boldsymbol{x}^2 + \sigma_t \epsilon, \alpha_t \boldsymbol{x}^3 + \sigma_t \epsilon \ldots \alpha_t \boldsymbol{x}^N + \sigma_t \epsilon \right] | \boldsymbol{c} \right) \tag{6}$$

where $0 < \alpha_t, \sigma_t < 1$ are scalars from chosen noising schedules(Karras et al. (2022); Song et al. (2021)). For any noising schedule, as $t$ increases, the signal from the original video is gradually shrinking and replaced by pure noise. In fact, the motion signal is even more easily getting buried in the noise than the overall appearance, explained below.

Consider the motion in a video, which usually only happens in a small area, on a single human figure or several moving objects, otherwise it will overwhelm the human visual cognition system. These clues lead us to claim that there is usually not much difference between $x^i$ and $x^{i+1}$ on the frame pixel values. For a normal video aligned with human perceptual preference, the change on frame pixels happen slowly to allow human eyes to capture meaningful sense.

In this context, define motion as the inter-frame difference $(x^{i+1} - x^i)$, the average numerical value for motion in a video is significantly smaller than the video itself.

$$\text{torch.mean}\big( \big[ ||x^2 - x^1||, ||x^3 - x^2||, \dots ||x^N - x^{N-1}|| \big] \big) \tag{7}$$
$$\lll \text{torch.mean}\big( \big[ ||x^1||, ||x^2||, ||x^3|| \dots ||x^N|| \big] \big)$$

Also after forward diffusing, the numerical value of motion drops linearly with $\alpha_t$.

$$(\alpha_t x^{i+1} + \sigma_t \epsilon) - (\alpha_t x^i + \sigma_t \epsilon) = \alpha_t (x^{i+1} - x^i) \tag{8}$$

The inherent small frame pixel change in videos and forward diffusing are two reasons identified by us leading to buried motion signal. Recheck the DMD loss in equation 3, where $G_\phi$ with the parameter $\phi$ to be updated lies within the input to the score function. This essentially prevents the student score model from perceiving the motion in generated videos, hence the evolving $\phi$ has no choice but to match the overall appearance, leading to buried motion signal in distillation.

## 4.2 Motion Score Matching

Given the current formulation of score matching and diffusion, polluting the generated video with Gaussian noise is unavoidable for computing the score function. Also, the teacher diffusion model trained on enormous data is a fixed pretrained model as in Equation 5. We cannot easily change the input variable due to the nature of score function. Hence, we have zero knowledge on

$$s\big( \big[ x^2 - x^1, x^3 - x^2 \dots x^N - x^{N-1} \big] \,|c \big), \tag{9}$$

because the base model is not trained on such data.

Under these constraints, changing the input to the score function will not provide extra information about motion. Turning to the output side and fortunately, the output from the teacher score model (Equation 5) still has a temporal dimension. Because the teacher model operates in a latent space encoded from video clips by 3D-VAE(Wan et al. (2025); Kong et al. (2024); Hong et al. (2022)) with an explicit temporal dimension structure, the output score value has a meaningful difference in temporal dimension that can be decoded into motion in final videos.

Here we introduce Motion Score (MS) on video score models. Given a score model $s$ defined on video clip $[x^1, x^2, x^3 \dots x^N]$, the motion score is defined as

$$s^{\text{motion}}\big( \big[ x^1, x^2, x^3 \dots x^N \big] \,|c \big) \triangleq s\big( \big[ x^1, x^2, x^3 \dots x^N \big] \,|c \big)_{[2:N]} \tag{10}$$
$$- s\big( \big[ x^1, x^2, x^3 \dots x^N \big] \,|c \big)_{[1:N-1]}$$

where $N$ is the number of total frames in a clip. Note that the indexing operator here only works on the temporal dimension. Motion Score is an explicit computation on the inter-frame difference over the score output which brings almost zero extra computational cost.

Consequently, the Motion Score Matching (MSM) loss is defined as:

$$\nabla_\phi \mathcal{L}_{\text{MSM}} \triangleq \mathbb{E}_t \big( \nabla_\phi \text{KL} \big( p_{\text{gen},t}^{\text{motion}} | p_{\text{data},t}^{\text{motion}} \big) \big) \tag{11}$$
$$\approx -\mathbb{E}_t \left( \int \big( s_{\text{data}}^{\text{motion}} \big( \Psi \left( G_\phi(\epsilon), t \right), t \big) - s_{\text{gen},\xi}^{\text{motion}} \big( \Psi \left( G_\phi(\epsilon), t \right), t \big) \big) \frac{dG_\phi(\epsilon)}{d\phi}, d\epsilon \right).$$

Compared to original Distribution Matching Distillation in Equation 3, this loss is an explicit enforcement on the generated motion rather than overall distribution matching. Due to the burial of motion signal in Equation 7 and 8, and the zero knowledge on Equation 9, enforcing the student generator to capture the subtle frame difference is one of the remaining choices to learn the motion explicitly.

In practice, Motion Score Matching(MSM) is recommended to be applied together with the current DMD loss. The overall Motion Enhanced Distribution Matching Distillation loss is

$$\nabla_\phi \mathcal{L}_{\text{DMD}} + \lambda \nabla_\phi \mathcal{L}_{\text{MSM}} \ . \tag{12}$$

This new loss is iteratively trained with the loss over the parameter of student score model the same as in DMD(Yin et al. (2024b)). The student score model (fake score model) is not affected by the introduction of MSM because the training objective of the student score model is to precisely estimate the distribution of the student generator, no matter the quality of generated videos. Also, it is difficult to augment the student score model on motion quality because we do not have knowledge on the ground truth motion score as in Equation 9.

Compared to the original DMD loss, this new loss introduces negligible extra computational cost. During forward pass, this loss adds only two tensor subtraction and one reduction operations, which are negligible for modern GPUs. During backward propagation, in addition to the original update on the student generator parameter $\phi$, the newly introduced MSM loss tweaks the updating direction towards favoring the motion distribution from favoring the original overall appearance distribution.

## 5 EXPERIMENTS

Distribution Matching Distillation was originally proposed to accelerate text-to-image generation in Yin et al. (2024b;a). This distillation scheme does not require simulating the full PF-ODE trajectory from the teacher diffusion model, saving a lot of inference time. This convenience makes DMD particularly suitable for distilling video generation models because they usually require more inference time compared to image generation models. Several works(Yin et al. (2025); Zhu et al. (2024); Lu et al. (2025); Shao et al. (2025); Luo et al. (2025)) have applied DMD in the distillation of video generation models. Among these works, only Yin et al. (2025) gives full training details by the time of our submission, which we conduct our experiments on. Note that the MSM loss in Equation 11 requires identical diffusing timestep $t$ for all frames, it is not yet applicable to the sequential distillation in Yin et al. (2025) where different level of noise is injected into different frames. This will be further addressed in Section 6.2.

### 5.1 EXPERIMENTAL SETUP

**Model Architecture** The base model to be distilled is Wan2.1-T2V(Wan et al. (2025)), an open-source bidirectional DiT(Peebles & Xie (2023)) model trained in a latent space encoded by a 3D VAE model from video clips. This teacher model is frozen during training, serving as $s_{\text{data}}$ in our loss. The student model $s_{\text{gen},\xi}$ is expected to do few-step generation, different from the one-step generation in Yin et al. (2024b). Hence, our student model shares exactly the same structure as the teacher as it also requires timestep input. Another DiT model, the score model for the student generator, also has the same architecture as the teacher. Both of the student generator and the student score model are initialized from the teacher checkpoint, which is critical for stable training and successful convergence as proven in Yin et al. (2024b) .

**Dataset** The original setup in Yin et al. (2025) involves proprietary datasets. Following the released code of this work, we use the open-source MixKit Datasets with 6484 videos from the Open-Sora Plan(Lin et al. (2024)). Due to the gap on data scale, our results may be different from those reported in Yin et al. (2025) . The other datasets mentioned in Yin et al. (2025) are too large to run on university facilities.

**Optimization** We use AdamW optimizer with learning rate $2 \times 10^{-6}$ and beta $(0.9, 0.999)$. Yin et al. (2025) mentioned that the training saturates around 1000 iterations on the MixKit dataset. But we observe that longer training generally provides better visual quality on the final generated videos. To fully show the advantage brought by our MSM, we entend the training of all experiments to 14000 iterations and report their results.

**Evaluation Metric** The evaluation on video motion quality is also an unsolved problem. In our context, a video with higher motion quality looks more realistic. And by realistic we mean the motion is considered normal in many aspects such as 3D geometry, physics, human bio-structure, lighting and graphics and many others. Also, we want our evaluation on motion to be independent from still appearance. Here we adopt the JACCARD and occulusion accuracy metric from TRAJAN(Allen et al. (2025)) . TRAJAN converts a video to tracking points first, and then calculate metrics only based on the points but not the videos. This completely eliminates the impact of appearance. Internally, this metric is a VAE model trained on lots of real videos. If a video contains unrealistic motion, it is more difficult to reconstruct using the pretrained VAE model. JACCARD measures

Table 1: Quantitative results on MSM

| Setting | CD-FVD ↓ | JACCARD ↓ | Occlusion Accuracy ↓ |
|---------|----------|-----------|---------------------|
| Stage 1 DMD | 171.81 | 0.6393 | 0.9258 |
| Stage 1 Ours | 143.12 | 0.6181 | 0.9128 |
| Stage 3 DMD | 211.45 | 0.5633 | 0.8849 |
| Stage 3 Ours | 187.62 | 0.5487 | 0.8815 |

how well the predicted visible points match the ground-truth visible points in each frame. Occlusion accuracy measures how often the model correctly predicts whether each point is visible or occluded. Normally, we would expect a higher score for both metrics for better quality. But TRAJAN also points out that both metrics are computed on the correctness of the reconstructed tracking points, which makes still videos score artificially high since they are trivially reconstructed. In our experiments, the occlusion accuracy is very high, above $90\%$. This means that the generated videos are mostly still, corresponding to our explanation in Section 4.1. Under this scenario, lower values on both TRAJAN and occlusion accuracy actually correspond to higher quality because higher values indicate still videos and motion signal loss.

In addition to these two new metrics, we also adopt traditional metrics for video quality measurement. We use Content Debiased Fréchet Video Distance(Ge et al. (2024)) to evaluate general video quality by computing the Fréchet Distance between the generated set of videos and the original dataset. Both contain 6484 videos with identical prompts.

## 5.2 EXPERIMENTAL RESULTS

There are three stages of distillation in Yin et al. (2024a) . In the first stage stage, the original 50-step video diffusion model is distilled into a 3 step model. The primary objective for this stage is to reduce the number of sampling steps, and efficiently generate a lot of samples as the training data for the second stage. We apply our MSM in this stage to compare with the base DMD methods.

The second stage is different, it does not distill on the number of sampling steps, but changes a bidirectional attention to sequential form to enable autoregressive generation. In this process, the timestep is different for each frame in a video, and our MSM cannot be applied due to reasons in Section 6.2 .

The third stage is working on the result from the second stage. After the model is converted to autoregressive, the third stage is working again on reducing the number of sampling steps. So this stage is almost identical to the first stage. It also tries to distill a 50-step diffusion model into a 3-step model, except that an extra step is needed in the end due to the autoregressive setup. We also apply MSM in this stage and keep our settings identical to the first stage.

Under the original DMD scheme, the motion signal is lost during distillation. The generated videos exhibit strange motion behaviors like suddenly appeared objects, weird object and camera movement. Our approach significantly mitigates this problem and the generated motion is much more fluent and vivid. We demonstrate these effects in Figure 1 and 2.

We report our quantitative results on Stage 1 and 3 in Table 1. The CD-FVD metric significantly drops in Stage 1 after using our approach. But in Stage 3, the drop on CD-FVD shrinks and the absolute value is still higher than Stage 1. This is due to the motion degradation in Stage 2 distillation. Our approach cannot be applied in Stage 2 yet, further explained in Section 6.2 . The input model for Stage 3 already suffers from poor motion quality, leading to degraded motion performance after Stage 3 distillation.

## 6 DISCUSSION AND FUTURE WORKS

For a video to have "realistic" motion, we mean it has to be correct on a lot of aspects including 3D geometry, physics, lighting and graphics, human bio-engineering etc. Normally these fields have their own specific methods, but it remains a great challenge to consider them all together

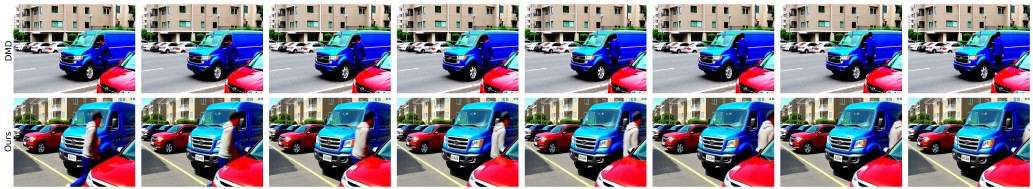

Figure 1: Different motion quality in generated videos. Both videos are generated from the same text prompt. Pay attention to the emergence of flower. In DMD, the flower emerged weirdly in front of the bowl. In ours, the flower is dropped down from some height. With our approach, the generated motion is much more reasonable.

Figure 2: Another example of improved motion quality. In DMD, the pedestrian irregularly morphs and turns around in front of the truck. In ours, the pedestrian walks by the vehicle normally.

when generating a video. However, the human have a simple intuition about "realistic" built upon extensive video data captured with our own eyes throughout our lives. This intuition inspires a distributional perspective on motion in videos. The current large video diffusion model is the closest model to this idea, but they are not specifically trained for motion. Motion Score as defined in Equation 10 is a compromise approach here to give a usable but fake score function on motion distribution, as the true score function requires training from scratch.

In this section, we first discuss potential alternative formats on the current motion score formulation. Then we discuss why the current MSM formulation is not applicable to autoregressive distillation. In the end, we reformulate the training objective of the current video diffusion models to give a dedicated motion score model, and point out the benefits of building such a foundational model.

## 6.1 OTHER FORMULATIONS ON MOTION SCORE

The definition of motion score in equation 10 can be further generalized into

$$s^{\mathrm{motion}}\left(\left[x^1, x^2, x^3 \ldots x^N\right] | c\right) \triangleq f\left(s\left(\left[x^1, x^2, x^3 \ldots x^N\right] | c\right)_{[2:N]}\right) \qquad (13)$$
$$- f\left(s\left(\left[x^1, x^2, x^3 \ldots x^N\right] | c\right)_{[1:N-1]}\right) ,$$

where $f$ is any feature extraction module defined over the output of the score function. This is usually done on real videos in optical flow with some pretrained motion extraction networks(Shi et al. (2023); Huang et al. (2022)). We tried some existing $f$, but they do not perform well. This is potentially due to the data modality of input, which is not real videos but score values on noised videos. It might be helpful if $f$ can be pretrained on score values. But this is also challenging since it needs to cope with different levels of injected noise.

## 6.2 EQUAL FRAME-WISE TIMESTEP

We explain why our formulation relies on identical frame-wise noising timestep. The key of our Motion Score definition in Equation 10 is to expose the inter-frame difference from the pretrained score model as much as possible. Then we defined MSM in Equation 11 to match this difference, using the motion score from the teacher as ground truth. If the frame-wise timestep is not identical, then the frame-wise score value difference is dominated by the choice of timestep $t$ and no longer reveals motion information. Unfortunately, unequal timestep arrangement for video generation is getting more and more popular to enable autoregressive video generation(Yin et al. (2025)) in order

to generate much longer videos(Shao et al. (2025)) . Our current motion score formulation cannot be extended to these methods yet.

## 6.3 DEDICATED MOTION SCORE MODEL

Our formulation of the motion score is severely limited by the pretrained video diffusion models. Given the importance of motion distribution and its connection to many other research areas, we propose to slightly change the training objective of current video diffusion models for a dedicated motion score model. The change is straightforward, current video diffusion models are trained to give

$$s\left(\left[\boldsymbol{x}^1, \boldsymbol{x}^2, \boldsymbol{x}^3 \ldots \boldsymbol{x}^N\right] | \boldsymbol{c}\right),$$

(14)

while we recommend to change to

$$\boldsymbol{s}^{\text{motion}}\left(\left[\boldsymbol{x}^2 - \boldsymbol{x}^1, \boldsymbol{x}^3 - \boldsymbol{x}^2 \ldots \boldsymbol{x}^N - \boldsymbol{x}^{N-1}\right] | \boldsymbol{c}\right).$$

(15)

If we add some previous frames in the condition $\boldsymbol{c}$,

$$\boldsymbol{s}^{\text{motion}}\left(\left[\boldsymbol{x}^2 - \boldsymbol{x}^1, \boldsymbol{x}^3 - \boldsymbol{x}^2 \ldots \boldsymbol{x}^N - \boldsymbol{x}^{N-1}\right] | \boldsymbol{c}, \boldsymbol{x}^1\right),$$

(16)

then the new training objective is very similar to the current objective of I2V models. This change would not require any extra data or model change, the only modification is a subtraction. By this simple modification, we now have the true score model on the motion distribution in videos $\boldsymbol{s}^{\text{motion}}$. And with two explicit score functions on motion $\boldsymbol{s}^{\text{motion}}$ and overall appearance $\boldsymbol{s}$, it is much simpler to adjust the final generation effect during inference only. For example, during generation, we can adjust the ratio of $\boldsymbol{s}$ and $\boldsymbol{s}^{\text{motion}}$ for different preferences on the final generated videos. This can be adjusted freely by users to meet their needs on motion quality.

Furthermore, if we change this training objective to a more general form,

$$\boldsymbol{s}^{\text{motion}}\left(\left[\boldsymbol{x}^N - \boldsymbol{x}^{N-1}, \boldsymbol{x}^{N+1} - \boldsymbol{x}^N, \ldots \boldsymbol{x}^{N+l} - \boldsymbol{x}^{N+l-1}\right] | \boldsymbol{c}, \left[\boldsymbol{x}^{N-k}, \boldsymbol{x}^{N-k+1}, \ldots \boldsymbol{x}^{N-1}\right]\right),$$

(17)

where $k$ is the size of the backward window and $l$ is the size of the forward window, then this model itself is an autoregressive video generation model working to predict the motion in the next $l$ frames based on previous $k$ frames. Now, because the motion in $\boldsymbol{x}^N - \boldsymbol{x}^{N-1}$ is estimated $l$ times, the final output can be a weighted sum of all these predictions. The coefficient distribution of this summarization can be chosen flexibly by users. Because the model itself is autoregressive, it no longer needs distillation to generate longer videos.

Such a dedicated motion score model can also serve as a better evaluation metric for accessing general motion quality in videos. If $\boldsymbol{s}^{\text{motion}}$ is adequately trained on abundant data, then it can give a relatively accurate estimate on the score value of a given arbitrary video. We can then integrate over the PF-ODE trajectory for that video to get an estimate on the probability of that video. Higher probability value means better motion quality.

## 7 SUMMARY

We point out the special temporal property of video data that is usually buried by the current distribution matching distillation scheme. This improper handling of video will lead to poor motion quality in generated videos, and we identified two reasons, motion buried by the forward diffusing coefficient and inherently small motion change compared to still pixel values. We proposed a new matching scheme, motion score matching, to address this issue and is proven to improve the quality of generated videos. Finally, we call for a change on the training objective of video diffusion models to build a generalized and dedicated motion score model, which can facilitate research in other areas such as 3D geometry, physics-aware generation, and graphics.

## 8 STATEMENT ON REPRODUCIBILITY

The core implementation of our motion score matching method is included in the supplementary materials as pseudocode. We will release the full project in the future.

## 9 LLM USAGE

LLMs are used to help polish the writing of this paper. They are not used in the development stage of the core idea of this paper. They are not used to retrieve related works, but are used to rephrase the summarization on related works for a more concise expression.

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
