# A  APPENDIX

We give the core implementation of our Motion Score Matching loss here. Compared to original DMD loss, only two tensor subtraction operations and one aggregation operation are introduced.

---

**Algorithm 1:** MotionMatchingLoss

```
# mu_real, mu_fake: denoising networks for real and
    fake distribution
# x: fake sample generated by our generator, in shape
    [B, T, C, H, W]
# min_dm_step, max_dm_step: timestep intervals for
    computing distribution matching loss
# bs: batch size
# N: total number of frames in video
# lambda: weighting factor

# random timesteps
timestep = randint(min_dm_step, max_dm_step, [bs])
noise = randn_like(x)

# Diffuse generated sample by injecting noise
# e.g. noise_x = x + noise * sigma_t (EDM)
noisy_x = forward_diffusion(x, noise, timestep)

# denoise using real and fake denoiser
with_grad_disabled():
    pred_fake_video = mu_fake(noisy_x, timestep)
    pred_real_video = mu_real(noisy_x, timestep)
    motion_fake = pred_fake_video[:,2:N,:,:,:] -
        pred_fake_video[:,1:N-1,:,:,:]
    motion_real = pred_real_video[:,2:N,:,:,:] -
        pred_real_video[:,1:N-1,:,:,:]

weighting_factor = abs(x - pred_real_image).mean(
    dim=[1, 2, 3], keepdim=True)
grad = (pred_fake_image - pred_real_image) /
    weighting_factor

# the loss that would enforce above grad
loss = 0.5 * mse_loss(x, stopgrad(x - grad)) + lambda*
    mse_loss(motion_fake - motion_real)
```

---