# OpenReview forum: "Motion Score Matching in Video Generation"
_ICLR.cc/2026/Conference — Submitted to ICLR 2026_

### Official Review · Reviewer_oerc · 2025-10-28

**Soundness:** 2
**Presentation:** 2
**Contribution:** 2
**Rating:** 4
**Confidence:** 4

**Summary:**

The paper shows that current video diffusion training may overlook temporal motion, harming video quality. It proposes Motion Score Matching to better model motion and improve generation. This approach also aims for broader applications in 3D, physics-aware, and graphics research.

**Strengths:**

The problem is clearly motivated, targeting a specific motion-related challenge in the popular DMD framework.

**Weaknesses:**

**Fundamental difference:**
  While the problem is clearly motivated, the proposed MSM approach mainly adds a regularization loss (eqs. 10 and 11); it is unclear what fundamental difference this introduces compared to the vanilla DMD loss.

**Comparison to DMD variants:**
  DMD has newer variants such as DMD2, and it would be helpful to clarify what aspects of motion modeling the MSM loss improves beyond these existing methods.

**Hyperparameter:**
  The weight $\\lambda$ in eq. 11 is not specified, and its impact on the results should be discussed.

**Evaluation metrics:**
  Including motion-specific metrics, such as FVMD [1], could better highlight the effect of the proposed method on motion quality.


Ref:\
[1] Liu, J., Qu, Y., Yan, Q., Zeng, X., Wang, L. and Liao, R., 2024. Fr'echet Video Motion Distance: A Metric for Evaluating Motion Consistency in Videos. arXiv preprint arXiv:2407.16124.

**Questions:**

Please see the [Weakness] section.

---

### Official Review · Reviewer_E2iG · 2025-10-28

**Soundness:** 2
**Presentation:** 2
**Contribution:** 2
**Rating:** 2
**Confidence:** 2

**Summary:**

The paper argues that in Distribution Matching Distillation (DMD) for video diffusion, motion cues are buried because adjacent frames differ only slightly and forward diffusion further shrinks those differences. It proposes Motion Score Matching (MSM): define a motion score as the temporal difference between the teacher’s score outputs of adjacent frames, and add a KL-matching term so the student aligns with the teacher in this motion-score space. MSM is combined with DMD (weighted by λ) and evaluated by distilling WAN2.1-T2V on MixKit. The paper reports CD-FVD improvements (and a re-interpretation of TRAJAN under mostly-static videos) in Stage-1 and smaller gains in Stage-3, while MSM is not applicable to Stage-2 (sequential distillation) because it requires identical timesteps for all frames.

**Strengths:**

1. Clear, plausible motivation for “motion burial.” Two contributing factors are articulated: (i) inter-frame signal is small relative to appearance; and (ii) forward diffusion scales inter-frame differences by \alpha_t, further diminishing motion. The accompanying derivations are straightforward and convincing.

2. Simple, low-overhead formulation. MSM amounts to a temporal difference over score outputs plus a KL term, combined with DMD via a scalar weight. It is easy to implement and essentially free at inference time.

3. Some quantitative signal. On MixKit, CD-FVD improves. The paper also discusses why its TRAJAN numbers are interpreted as lower-is-better in mostly-static settings.

**Weaknesses:**

## major
1. Insufficient qualitative evidence and visualization protocol

	•	Current presentation relies on only a couple of examples, which is not enough to substantiate claims of noticeably improved motion.

	•	Adopt a systematic side-by-side protocol: same prompts, same seeds, same frame counts; frame-aligned comparisons; zoom-ins / slow-motion snippets to expose motion details.

	•	Include an intuitive “motion burial” visualization (e.g., illustrating per-frame score/flow magnitude over time) to make the motivation directly observable.

	•	Consider a human preference study (pairwise votes/CMOS) to complement metric-based evidence.

2. Limited quantitative scope and metric framing

	•	Experiments are only on MixKit (~6.5k clips); add at least one more public dataset (e.g., WebVid, UCF-101, MSR-VTT) to test generality.

	•	The TRAJAN lower-is-better re-interpretation under mostly static videos introduces ambiguity; pair it with motion-oriented diagnostics (e.g., optical-flow consistency, speed/acceleration distributions, trajectory smoothness) and standard CD-FVD/FVD to reduce reliance on any single interpretation.

	•	Report results consistently across multiple metrics and summarize trade-offs (appearance vs. motion) to clarify the operating regime.

3. Missing ablations, robustness, and comparative baselines

	•	Provide sensitivity analyses: MSM weight \lambda, timestep schedules, sampling steps/video length.

	•	Test different teachers (e.g., CogVideoX) and vary video duration/frame rate to probe robustness.

	•	Add comparisons to alternative motion-enhancing distillation or motion/appearance-disentangling methods to position MSM within the landscape.

	•	If feasible, include cross-dataset generalization as part of robustness (train on one, evaluate on another).

## minor

4. Reproducibility details are thin. While the WAN2.1-T2V teacher and the student setup are outlined, the implementation appears to rely on external code with a promise of future release. Practical details—compute budget for training to 14k iters, data cleaning/curation, and other training specifics—are not fully documented.

5. Scope/structure feels imbalanced. A sizeable portion of the paper is devoted to Discussion/Future Work, while central empirical questions remain under-explored. The contribution reads as an incremental, practical add-on that would benefit from broader experiments and stronger visual evidence to close the empirical loop.

6. Ablations and robustness (if not already covered above). It would be helpful to report sensitivities to \lambda, timestep sampling schedules, frame rate/video length, and different teachers (e.g., CogVideoX), and to show generalization on at least one additional public dataset. This would clarify robustness and the method’s operating envelope.

**Questions:**

See Weaknesses.

---

### Official Review · Reviewer_eue4 · 2025-10-30

**Soundness:** 2
**Presentation:** 2
**Contribution:** 2
**Rating:** 2
**Confidence:** 4

**Summary:**

this paper is trying to tackle the problem of video diffusion models producing weird or janky motion. The authors argue that current distillation methods (DMD) tend to gloss over subtle movements, messing up the final video.

The main contribution here is "Motion Score Matching" (MSM). It's a new loss function that, during distillation, explicitly tells the student model to copy the motion distribution of the teacher model, not just the final look of the frames. It's calculated by looking at the difference in score outputs between adjacent frames. The paper gives a bit of theory for why this should work and claims it adds almost no extra compute cost.

You've tested this using metrics like CD-FVD and TRAJAN, and included some visual examples.

**Strengths:**

- Good Experimental Proof: The numbers, especially in Table 1 (p. 7), look pretty good. It shows MSM beating standard DMD on motion-specific stuff (JACCARD, occlusion) and also on the general (CD-FVD) metric. This seems most helpful in the early stages of distillation.

- Honesty About Limits: I appreciate that you were upfront about what MSM can't do, like how it's currently not usable for sequential/unequal-timestep distillation (Section 6.2).

- Clear Setup: You did a good job laying out the training schedule, optimizer, dataset, etc. (Section 5.1). It seems like someone could reasonably reproduce your work.

**Weaknesses:**

- The "Why" is a Bit Weak:

You introduce the motion score as just the difference between adjacent frames (Eq. 10), but the paper doesn't really give a rock-solid, formal proof for why this is the optimal way to do it, especially when you're dealing with noisy frames. The argument feels more intuitive than rigorous (Section 4.2). I was also looking for some ablations here—like, did you try different ways of defining "motion"? What if you weighted the loss differently?

- Doesn't Work on SOTA Pipelines:

This is a big one. You state (in 5.2 and 6.2) that MSM can't be used when different noise timesteps are applied to different frames. Since that's how a lot of modern, state-of-the-art video models work, this severely limits the impact and real-world usefulness of your paper. Just pointing it out as a limitation isn't quite enough; it's a critical weakness.

- Are the CD-FVD Gains Just from Longer Training?

Looking at Table 1, the improvement on CD-FVD is... okay, it's moderate. But more importantly, you trained your MSM model for 14,000 iterations versus the baseline's 1,000. That's 14x longer. It's impossible to tell if the improvement is from your method or just from the massive difference in training time. We need an apples-to-apples comparison.

- Evaluation is on Simple Stuff:

The figures you show (Figs 1, 2) are all pretty short, simple, single actions. There's no evidence here that MSM helps with more complex, long-term video challenges, like tracking multiple objects, maintaining consistent motion over dozens of frames, or handling complex trajectories.

**Questions:**

- Other Motion Operators? Did you experiment with any other ways to formulate the motion score, beyond just a simple frame difference? Maybe something like higher-order time derivatives, or using optical flow (which you hint at in 6.1)? Does the choice of operator make a big difference?

- Longer or Busier Videos? You focused on short, single-action clips. Have you looked at how MSM performs on longer sequences, or in scenes with multiple moving objects and occlusions? Does the motion realism gain still hold up, or does it fade?

- Any Fixes for the Unequal-Timestep Problem? In 6.2 you mention MSM can't be used in sequential distillation. Can you suggest any research directions or ideas on how to get around this? This seems crucial for the method to be widely adopted.

- Ablation on Training Time? Given the 14,000 vs. 1,000 iteration difference, could you run a controlled experiment to isolate the effect of MSM from the effect of just training longer?

- Can your provide more video generation demonstration?

---

### Official Review · Reviewer_476N · 2025-10-31

**Soundness:** 1
**Presentation:** 1
**Contribution:** 1
**Rating:** 2
**Confidence:** 5

**Summary:**

The paper proposes to introduce a motion score diffusion for video generation. The paper demonstrates that preserving the motion signal is vital for video generation quality.

**Strengths:**

The motivation of introducing motion clues for video generation is interesting.

Identifying two causes of buried motion signal is interesting.

**Weaknesses:**

1. In the abstract, “Current distillation methods prefer to match the style first, since it takes up most of the numerical significance. Such a distillation scheme will only create poorly generated motions,”, please add high-level intuition or explanation.

2. Line 34, what is PF-ODE? The term appears without explanations. ODE is a common terminology, but PF-ODE (presumably,  Probability Flow (PF) ODE) is not.

3. Eq. 7 uses implementation-specific notation (‘torch.mean’). Please replace framework code with standard mathematical notation.

4. Motion score” is essentially a finite-difference along the temporal index of the teacher’s score (Eq. 10), i.e., a simple frame-to-frame subtraction of the score output. Beyond the intuition that motion signals are small (Eqs. 7–8), there’s little new theory or a principled derivation of why this particular differencing is the right statistic to distill—or how it relates to the teacher’s underlying SDE/PF-ODE. Claims like “prove that preserving motion is vital” are stated but not accompanied by a formal theorem/proof.

5. Also, prior works (e.g., [1,2]) have leveraged explicit optical flow for motion control/consistency with clear interpretability, whereas “motion score” is a more implicit proxy and should be positioned relative to these flow-based lines.

6. The method requires identical timesteps across frames; authors explicitly say it “cannot be applied” to modern autoregressive/unequal-timestep pipelines (Stage 2 and many current T2V systems) (Sec. 5 & Sec. 6.2). That sharply narrows the scope to a subset of bidirectional, equal-noise settings.

7. The paper adopts TRAJAN’s JACCARD & Occlusion Accuracy but then inverts the interpretation, arguing that lower is better because still videos score “artificially high.” This runs counter to metric semantics and risks confusion and cherry-picking. Human studies or motion realism judgments are absent; temporal consistency or optical-flow stability metrics are not reported as primary evidence. Only CD-FVD is added, with mixed gains.

8. Adding MSM could harm spatial fidelity or style. There’s no ablation over the weight $\lambda$ in Eq. 12, no FID/LPIPS-like appearance trade-off curves, and no user preference study quantifying motion realism vs. appearance degradation.

9. Baselines are too narrow. Comparisons focus mainly on DMD with/without MSM. Other motion-aware distillation/regularization baselines (including those the paper itself discusses in related work) aren’t discussed under the same setup, so it’s hard to tell if MSM is better than existing motion-appearance disentangling strategies.



[1]. Nam H, Kim J, Lee D, et al. Optical-flow guided prompt optimization for coherent video generation[C]//Proceedings of the Computer Vision and Pattern Recognition Conference. 2025: 7837-7846.

[2]. Geng D, Herrmann C, Hur J, et al. Motion prompting: Controlling video generation with motion trajectories[C]//Proceedings of the Computer Vision and Pattern Recognition Conference. 2025: 1-12.

**Questions:**

See my weakness.

---

### Meta-Review · Area_Chair_LP8f · 2025-12-24

**Summary:**

This paper introduces Motion Score Matching (MSM) to improve motion realism in video distillation by enforcing consistency between the student and teacher models' temporal score differences. The authors aim to solve the motion burial problem, where subtle motion cues are often lost during standard distribution matching distillation.
Reviewers appreciated the clear motivation of the work, agreeing that preserving motion quality is a critical challenge in video generation. They also noted that the problem formulation regarding the loss of motion signals in current methods is intuitive.

The reviewers identified several critical flaws that lower the paper's ratings:
1. **Lack of Theoretical Grounding**: Multiple reviewers pointed out that the proposed motion score, which is defined simply as the finite difference between adjacent frames, is a heuristic without a rigorous mathematical derivation. There is no formal proof connecting this differencing operation to the underlying SDE or Probability Flow ODE, making the theoretical contribution weak. (476N, eue4)
2.** Limited Practical Applicability**: A major technical limitation is that the method requires identical noise timesteps across frames, which cannot be applied to modern autoregressive/unequal-timestep pipelines, which narrows its scope and impact. (476N, eue4)
3. **Unfair Experimental Comparisons**: The experimental validation was heavily criticized for lacking fair comparisons. Specifically, the proposed method was trained for 14,000 iterations while the baseline was only trained for 1,000. This disparity makes it impossible to attribute performance gains to the method rather than simply longer training. (eue4)
4. **Questionable Metrics and Baselines**: The evaluation was described as insufficient. Reviewers noted that the interpretation of metrics like TRAJAN (claiming lower is better for static scenes) seemed to contradict standard usage. Furthermore, the baselines were too narrow, lacking comparisons to other motion-aware distillation techniques. (476N, E2iG, oerc)

In summary, this paper was reviewed by four experts in the field. The recommendations are uniformly negative (Scores: 2, 2, 2, 4). The reviewers reached a consensus that the method lacks theoretical depth, practical utility for modern models, and fair empirical validation. The authors did not submit a rebuttal to address these concerns.

**Reviewer Concerns:**

**Well addressed**:

None.

**Partly addressed**:

None.

**Unsolved**:
1. Lack of Theoretical Grounding  (476N, eue4)
2. Limited Practical Applicability (476N, eue4)
3. Unfair Experimental Comparisons (eue4)
4. Questionable Metrics and Baselines (476N, E2iG, oerc)

**Reviewer Scores:**

Since there are no rebuttal, the reviewers' scores would remain.

---

### Decision · Program_Chairs · 2026-01-26

Reject